# Biomarkers of sickle cell nephropathy in Senegal

El Hadji Malick Ndour[1,2]*, Khuthala Mnika[3,4], Fatou Guèye Tall[1,2], Moussa Seck[5], Indou Dème Ly[2], Victoria Nembaware[3], Gaston Kuzamunu Mazandu[3], Hélène Ange Thérèse Sagna Bassène[2], Rokhaya Dione[2], Aliou Abdoulaye Ndongo[6], Jean Pascal Demba Diop[7], Nènè Oumou Kesso Barry[1], Moustapha Djité[1], Rokhaya Ndiaye Diallo[7], Papa Madiève Guèye[1], Saliou Diop[5], Ibrahima Diagne[8], Aynina Cissé[1], Ambroise Wonkam[3,9], Philomène Lopez Sall[1,2]

1 Department of Pharmaceutical Biochemistry, Faculty of Medicine, Pharmacy and Dentistry, Cheikh Anta Diop University, Dakar, Senegal, 2 Albert Royer National University Hospital of Children, Dakar, Senegal, 3 Division of Human Genetics, Department of Pathology, Faculty of Health Sciences, University of Cape Town, Cape Town, South Africa, 4 Division of Human Genetics, National Health Laboratory Service, and School of Pathology, Faculty of Health Sciences, University of the Witwatersrand, Johannesburg, South Africa, 5 National Center of Blood Transfusion, Dakar, Senegal, 6 Department of Pediatrics, Dantec National University Hospital, Dakar, Senegal, 7 Department of Human Genetics, Faculty of Medicine, Pharmacy and Dentistry, Cheikh Anta Diop University, Dakar, Senegal, 8 Department of Pediatrics, Faculty of Health Sciences, Gaston Berger University, Saint-Louis, Senegal, 9 McKusick-Nathans Institute and Department of Genetic Medicine, Johns Hopkins University School of Medicine, Baltimore, Maryland, United States of America

* elhadjimalickndour@yahoo.fr

**Data Availability Statement:** All relevant data are within the paper and its Supporting Information files.

**Funding:** The clinical chemistry experiments of the study were in part funded by The African Center of

## Abstract

Sickle cell anemia (SCA) is caused by a single point variation in the β-globin gene (*HBB*): c.20A> T (p.Glu7Val), in homozygous state. SCA is characterized by sickling of red blood cells in small blood vessels which leads to a range of multiorgan complications, including kidney dysfunction. This case-control study aims at identifying sickle cell nephropathy biomarkers in a group of patients living with SCA from Senegal. A total of 163 patients living with SCA and 177 ethnic matched controls were investigated. Biological phenotyping included evaluation of glycemia, glucosuria, albuminuria, proteinuria, tubular proteinuria, serum creatinine, urine creatinine, urine specific gravity and glomerular filtration rate. Descriptive statistics of biomarkers were performed using the χ2 –test, with the significance level set at p<0.05. Patients living with SCA had a median age of 20 years (range 4 to 57) with a female sex frequency of 53.21%. The median age of the control participants was 29 years (range: 4–77) with a female sex frequency of 66.09%. The following proportions of abnormal biological indices were observed in SCA patients versus (vs.) controls, as follows: hyposthenuria: 35.3% vs.5.2% (p<0.001); glomerular hyperfiltration: 47.66%vs.19.75% (p<0.001), renal insufficiency: 5.47%vs.3.82% (p = 0.182); microalbuminuria: 42.38%vs.5.78% (p<0.001); proteinuria: 39.33%vs.4.62% (p<0.001); tubular proteinuria: 40.97%vs.4.73% (p<0.001) and microglucosuria: 22.5%vs.5.1% (p<0.001). This study shows a relatively high proportion of SCA nephropathy among patients living with SCA in Senegal. Microglucosuria, proteinuria, tubular proteinuria, microalbuminuria, hyposthenuria and glomerular hyperfiltration are the most prevalent biomarkers of nephropathy in this group of Senegalese patients with SCA.

Excellence for Maternal and Child Health «Centre d'Excellence Africain pour la Santé de la Mère et de l'Enfant (CEA-SAMEF, http://ceasamef.sn, No 000099/2018/JCM/KND) », Cheikh Anta Diop University of Dakar, Senegal. The molecular experiments of the study were funded by SADaCC (https://www.sickleinafrica.org) at the Division of Human Genetics, Faculty for Health Sciences, University of Cape Town, South Africa. The funders had no role in study design, data collection and analysis, decision to publish, or preparation of the manuscript.

**Competing interests:** The authors have declared that no competing interests exist.

**Abbreviations:** SCD, ; NM_000518.**5**:c.20A>T, substitution of A to T at nucleotide position 20 of the complementary DNA; A, adenine; T, Thymine; NP_000509.**1**, p.Glu7Val: replacement of glutamic acid by valine at position 7 of the protein (β-globin chain); Glu, Glutamic acid; Val, Valine; SCA, Sickle cell anemia; SNP, single nucleotide polymorphism; C, Cytosine; GFR, glomerular filtration rate; USG, urine specific gravity; ESRD, End-stage renal disease; SS, SCA patients without diabetes; CNTS, National Blood Transfusion Center; USAD, Ambulatory Care Unit for Children and Adolescents with Sickle Cell Disease; CHEAR, Albert Royer National University Children's Hospital; CKD-EPI, Chronic Kidney Disease–EPIdemiology; UPCR, urinary protein to creatinine ratio; UACR, urinary albumin to creatinine ratio; UGCR, urinary glucose to creatinine ratio; GHF, Glomerular hyperfiltration; RFLP, restriction fragment length polymorphism; PCR, polymerase chain reaction; OR, odds ratio; USA, United State of America; mOsm/kgH$_2$O, urine osmolality; PT, proximal tubule.

# Introduction

Sickle cell disease (SCD) refers to a collection of inherited blood disorders that feature the propensity for erythrocytes to change into crescent or so-called sickle shapes [1]. It is an hemoglobinopathy with autosomal recessive transmission caused by a single nucleotide substitution NM_000518.**5**:c.20A>T of the β-globin gene (*HBB*-rs334), located on the short arm of chromosome 11 (11p15.4) [2, 3]. The variation results in an amino-acid replacement NP_000509.**1**: p.Glu7Val of the β-globin chain of tetrameric hemoglobin (α$_2$β$_2$) in adults NM_000518.**5** (HBB):c.20A>T(p.Glu7Val) [1, 2]. Sickle cell anemia (SCA) refers to the disease which results from the homozygous expression of the β$^S$ allele (β$^S$/β$^S$ genotype) [4].

SCD is considered to be the most common monogenic disease in the world [5]. It is estimated that 305,800 children in the world, of whom 85% in sub-Saharan Africa, are born with SCD each year, and this number could reach 404,200 in 2050 [5]. Senegal is a country in sub-Saharan West Africa with a population of approximately 16,209,125 of which up to 2% are SCD patients [6, 7]. Three centers specializing in lifelong medical treatment for SCD patients have been established, but there is no universal newborns screening yet and very few patients are exposed to hydroxycarbamide. There is no universal medical insurance coverage and care for SCD patients is thus paid for by family members in this developing country where poverty affects from 24.9% of the population living in Dakar the capital to 77.5% of the rural population of the region of Kolda [8]. Therefore, the financial burden is heavy and the necessary medical care is often out of reach, and patients suffer from multiple SCD complications albeit the vast majority of the patients show the Senegal haplotype (*XmnI-rs7482144*) bearing the C>T single nucleotide polymorphism (SNP) at position –158 of the $^G$γ- globin gene (*HBG2*:g.-158C>T or NM_000184.**2**(HBG2):c.-211C>T) which is associated with higher fetal hemoglobin (HbF) levels known to result in a less severe clinical expression of SCD [3, 4].

Patients living with SCD may exhibit multiple organ damage including kidney dysfunction that may be structural and/or functional [9]. These glomerular and/or tubular renal damages prove to be age dependent [9]. In early childhood, kidney dysfunction is mainly glomerular hyperfiltration characterized by increased glomerular filtration rate (GFR), and loss of urinary concentration ability through the Henle's loop of juxtaglomerular nephrons resulting in hyposthenuria (*i.e.* a decrease in urine specific gravity (USG) is described [9]. In childhood, albuminuria stage A2 (microalbuminuria) is the most observed kidney dysfunction in SCD patients [9]. In adulthood, albuminuria stage A3 (macroalbuminuria) begins to develop and may be associated with renal insufficiency (*i.e.* a decrease in GFR) [9]. End-stage renal disease (ESRD) requiring hemodialysis and / or kidney transplantation occurs in 4–18% of SCD patients [10]. The average survival time after the onset of ESRD is 4 years, and 40% of SCD patients die after 20 months of dialysis [11].

However, in Africa, particularly in Senegal, only few studies have been carried out on renal manifestations of SCD, mainly on albuminuria and glomerular hyperfiltration and the available studies have been centered in patients with very low proportion of Senegal haplotype [12–15]. It is anticipated that early diagnosis of kidney dysfunction would allow early therapeutic intervention that could delay the onset of ESRD and increase the life expectancy of SCD patients [13].

Thus, the main objective of this study is to identify biomarkers of nephropathy that could be used for anticipatory guidance, and affordable routine clinical assessment of Senegalese patients living with sickle cell anemia (SCA).

# Results

Among 394 recruited subjects 164 (41.62%) were *HBB*-rs334 T (sickle mutation) in homozygous state, 49 (12.44%) in heterozygous state and 181 (45.94%) were *HBB*-rs334 A in

**Table 1. Description of anthropometric and biochemical parameters of controls.**

|  | Controls without sickle cell anemia and diabetes (n = 177) | | |
|---|---|---|---|
|  | Median (min—max) | 5 – 95th percentiles | Observations |
| Age, years | 29 (4–77) | 10–61 | 168 |
| Female sex, % (n) | 66.09% (115) | XXX | 174 |
| BMI, kg/m² | 22.38 (10.56–49.61) | 13.52–34.02 | 162 |
| SBP, mmHg | 120 (90–160) | 100–140 | 100 |
| DBP, mmHg | 70 (50–90) | 60–80 | 100 |
| Hb, g/dl | 13.3 (7.4–18.2) | 10.2–16.2 | 170 |
| Glycemia, mg/dl | 84 (70–110) | 73–98 | 170 |
| BUN, mg/dl | 8 (2.5–31.5) | 5–14 | 162 |
| Serum creatinine, mg/dl | 0.78 (0.23–7.36) | 0.38–1.4 | 166 |
| Urine creatinine, mg/dl | 227 (22–886) | 76–486 | 174 |
| GFR, ml/min/1.73m² | 114.41 (10.85–200.37) | 68.28–160.60 | 157 |
| Albuminuria (UACR, mg/g) | 6.52 (1.28–139.07) | 2.62–32.43 | 173 |
| Proteinuria (UPCR, mg/g) | 44.74 (9.21–530.27) | 15.07–181.48 | 173 |
| Glucosuria (UGCR, mg/g) | 0 (0–52) | 0–20 | 176 |
| USG | 1.020 (1.004–1.035) | 1.010–1.028 | 154 |

SBP, mmHg. DBP, mmHg. Hb, g/dl (x 0.6206 mmol/l). Glycemia, mg/dl (x 0.0555 mmol/l). BUN, mg/dl (x 0.357 mmol/l). Serum creatinine, mg/dl (x 88.4 μmol/l).
Urine creatinine, mg/dl (x 88.4 μmol/l). UACR, mg/g (x 0.113 mg/mmol). UPCR, mg/g (x 0.113 mg/mmol). UGCR, mg/g (x 0.625 μmol/mmol).
Min: minimum. Max: maximum. BMI: body mass index. SBP: systolic blood pressure. DBP: diastolic blood pressure. Hb: hemoglobin. BUN: blood urea nitrogen. GFR:
glomerular filtration rate is determined using Schwartz formula in children and adolescents and CKD-EPI formula in adults. UPCR: urinary protein /creatinine ratio.
UACR: urinary albumin/creatinine ratio. UGCR: urinary glucose/creatinine ratio. USG: urine specific gravity.

homozygous state. One of the 164 βS/βS patients and four of the 181 βA/βA one were excluded from the study because of hyperglycemia as well as the 49 βS/βA subjects. A total of 163 SCA patients and 177 ethnic matched controls without diabetes were therefore included in our series. Among the selected SCA patients 79 (63.71%) were Senegal haplotype and 45 (27.61%) were matched in age and sex with 45 of the 177 controls. Tables 1 and 2 summarize anthropometric and biochemical characteristics of the study participants. The median age of controls was 29 years [range 4–77]. Women were more represented than men in this control group with a frequency of 66.09% (n = 115) "Table 1". The control group consisted of 19.05% (n = 32) children and 80.95% (n = 136) adults. SCA patients had a median age of 20 years [range 4–57] with a female sex frequency of 53.21% (n = 83) in the SCA group "Table 2". SCA patients group was composed of 52.08% (n = 75) children and 47.92% (n = 69) adults.

The median serum creatinine observed in 166 controls was 0.78 mg/dl (68.95 μmol/l) [range 0.23–7.36 (20.33–650.62)] with 90% (n = 150) of this group having serum creatinine between 0.38 (33.59 μmol/l) and 1.4 mg/dl (123.76 μmol/l) "Table 1". The reference intervals of serum creatinine obtained from the controls of our series are therefore 0.78 mg/dl (68.95 μmol/l) [range 0.38–1.4 (33.59–123.76)] "Table 1". The mean serum creatinine level was significantly reduced in SCA compared to controls [0.59±0.23 (52.15±20.33) versus 0.86±0.60 (76.02±53.04) mg/dl (μmol/l); p < 0.001] "Table 3". The median glomerular filtration rate (GFR) was 136.96 ml/min /1.73m² [range 58.51–196.80] for SCA patients and 114.41 ml/min/1.73m² [range 68.28–160.60] for controls "Tables 1 and 2". The median urinary specific gravity (USG) was 1.012 [range: 1.007–1.020] and USG was ≤ 1.015 in 90.2% (n = 138) of SCA patients "Tables 2 and 4". In controls, the 5th percentile of USG was 1.010 "Table 1". The 95th percentile for albuminuria, proteinuria and glucosuria were respectively 32.43 mg/g (3.66 mg/mmol), 181.48 mg/ g (20.51 mg/mmol) and 20 mg/g (12.5 μmol/mmol) in the control group "Table 1".

**Table 2. Description of anthropometric and biochemical parameters of sickle cell anemia patients.**

| | Sickle cell anemia patients without diabetes (n = 163) | | |
|---|---|---|---|
| | Median (min—max) | 5 – 95th percentiles | Observations |
| Age, years | 20 (4–57) | 6–38 | 144 |
| Female sex, % (n) | 53.21% (83) | XXX | 156 |
| BMI, kg/m$^2$ | 18 (11.24–33.71) | 13.15–26.04 | 128 |
| SBP, mmHg | 110 (80–140) | 90–140 | 101 |
| DBP, mmHg | 70 (40–110) | 50–80 | 101 |
| Hb, g/dl | 8.4 (5.1–11.6) | 6.6–11 | 148 |
| Glycemia, mg/dl | 85 (60–108) | 71–101 | 152 |
| BUN, mg/dl | 6.5 (2.5–54.5) | 4–10.5 | 143 |
| Serum creatinine, mg/dl | 0.57 (0.13–1.58) | 0.27–0.99 | 151 |
| Urine creatinine, mg/dl | 88 (16–679) | 37–233 | 152 |
| GFR, ml/min/1.73m$^2$ | 136.96 (38.31–407.84) | 58.51–196.80 | 129 |
| Albuminuria (UACR, mg/g) | 25.67 (2.64–328.65) | 6.86–122.63 | 151 |
| Proteinuria (UPCR, mg/g) | 156.67 (17.24–2957.84) | 29.41–1388.76 | 150 |
| Glucosuria (UGCR, mg/g) | 5 (0–7370) | 0–193 | 151 |
| USG | 1.012 (1.001–1.025) | 1.007–1.020 | 153 |

SBP, mmHg. DBP, mmHg. Hb, g/dl (x 0.6206 mmol/l). Glycemia, mg/dl (x 0.0555 mmol/l). BUN, mg/dl (x 0.357 mmol/l). Serum creatinine, mg/dl (x 88.4 µmol/l). Urine creatinine, mg/dl (x 88.4 µmol/l). UACR, mg/g (x 0.113 mg/mmol). UPCR, mg/g (x 0.113 mg/mmol). RGCU, mg/g (x 0.625 µmol/mmol).

Min: minimum. Max: maximum. BMI: body mass index. SBP: systolic blood pressure. DBP: diastolic blood pressure. Hb: hemoglobin. BUN: blood urea nitrogen. GFR: glomerular filtration rate is determined using Schwartz formula in children and adolescents and CKD-EPI formula in adults. UPCR: urinary protein /creatinine ratio. UACR: urinary albumin/creatinine ratio. UGCR: urinary glucose/creatinine ratio. USG: urine specific gravity.

Comparison of prevalence of biological indices abnormalities between SCA patients and unmatched controls showed that SCA was associated with kidney dysfunction "Table 4". The following proportions of abnormal biological indices of kidney dysfunction were observed in SCA patients versus controls, as follows: hyposthenuria: 35.3% versus 5.2% (p < 0.001);

**Table 3. Comparison of means of anthropometric and biochemical parameters between sickle cell anemia patients and unmatched controls and then age- and sex-matched controls.**

| | SS (n = 163) | Controls (n = 177) | | SS (n = 45) | Controls (n = 45) | |
|---|---|---|---|---|---|---|
| | Mean±SD or % (n) | Mean±SD or % (n) | p-value | Mean±SD or % (n) | Mean±SD or % (n) | p-value |
| Age, years | 20.4±10.2 | 32.1±15.3 | < 0.001 | 21.4±10.4 | 21.4±10.4 | 1 |
| Female sex, % (n) | 53.21%(83) | 66.09%(115) | 0.017 | 64.44%(29) | 64.44%(29) | 1 |
| BMI, kg/m$^2$ | 19.1± 6.8 | 22.9±6.8 | < 0.001 | 19.6±7.5 | 21,5±6.8 | 0.079 |
| SBP, mmHg | 112±13.0 | 117±13.5 | 0.0041 | 112±11.5 | 118±10.1 | 0.076 |
| DBP, mmHg | 69±11.2 | 72.5±8.0 | 0.0085 | 72.5±10.6 | 73.5±7.9 | 0.396 |
| Hb, g/dl | 8.57±1.47 | 13.38±1.8 | < 0.001 | 8.69±1.43 | 13.05±1.87 | < 0.001 |
| Glycemia, mg/dl | 85±9 | 86±8 | 0.71 | 85±7 | 86±10 | 0.447 |
| BUN, mg/dl | 7±4.5 | 9±3.5 | < 0.001 | 6.5±2 | 8.5±2.5 | < 0.001 |
| Serum creatinine, mg/dl | 0.59±0.23 | 0.86±0.60 | < 0.001 | 0.58±0.21 | 0.75±0.32 | 0.0085 |

SBP, mmHg. DBP, mmHg. Hb, g/dl (x 0.6206 mmol/l). Glycemia, mg/dl (x 0.0555 mmol/l). BUN, mg/dl (x 0.357 mmol/l). Serum creatinine, mg/dl (x 88.4 µmol/l). SS: sickle cell anemia patients without diabetes. Controls: subjects without sickle cell anemia and diabetes. BMI: body mass index. SBP: systolic blood pressure. DBP: diastolic blood pressure. Hb: hemoglobin. BUN: blood urea nitrogen.

**Table 4. Comparison of disturbances of nephropathy biomarkers between sickle cell anemia patients and controls.**

| | SS (n = 163) | Controls (n = 177) | | |
|---|---|---|---|---|
| | % (n) | % (n) | p-value | Odds ratio |
| Hyposthenuria | 35.3% (54) | 5.2% (8) | < 0.001 | 9.95 [4.43–25.12] |
| Glomerular hyperfiltration | 47.66% (61) | 19.75% (31) | < 0.001 | 3.64 [2–6.64] |
| Normal glomerular filtration | 35.94% (46) | 54.14% (85) | NA | NA |
| Glomerular hypofiltration | 10.94% (14) | 22.29% (35) | 0.407 | NA |
| Renal insufficiency | 5.47% (7) | 3.82% (6) | 0.182 | NA |
| Microalbuminuria | 42.38% (64) | 5.78% (10) | < 0.001 | 11.99 [5.71–27.33] |
| Proteinuria | 39.33% (59) | 4.62% (8) | < 0.001 | 13.37 [5.97–33.59] |
| Glomerular proteinuria | 0 | 0 | NA | NA |
| Tubular proteinuria | 40.97% (59) | 4.73% (8) | < 0.001 | 13.97 [6.22–35.15] |
| Microglucosuria | 22.5% (34) | 5.1% (9) | < 0.001 | 5.39 [2.41–13.21] |

SS: sickle cell anemia patients without diabetes. Controls: subjects without sickle cell anemia, sickle cell trait and diabetes. NA: not applicable.

glomerular hyperfiltration: 47.66% versus 19.75% (p < 0.001), glomerular hypofiltration: 10.94% versus 22.29% (p = 0.407); renal insufficiency: 5.47% versus 3.82% (p = 0.182); microalbuminuria (albuminuria stage A2): 42.38% versus 5.78% (p<0.001); proteinuria: 39.33% versus 4.62% (p < 0.001); tubular proteinuria: 40.97% versus 4.73% (p < 0.001) and microglucosuria: 22.5% versus 5.1% (p < 0.001) "Table 4". Glomerular proteinuria was absent in both groups "Table 4". Macroalbuminuria (albuminuria stage A3) was only observed in one ten-year-old female child in SCA patients group. Furthermore, SCA appears as a risk factor of kidney dysfunction biomarkers disruption if the odds ratio (OR) was taken into account "Table 4". The association between SCA and kidney dysfunction was confirmed by comparing kidney dysfunction biomarkers between SCA patients and controls matched in age and sex "Table 5".

## Discussion

To our knowledge, this study is the first to investigate kidney dysfunction in a group of patients living with SCA from Senegal. It has revealed a relatively high proportion of patients

**Table 5. Comparison of disturbances of nephropathy biomarkers between sickle cell anemia patients and age- and sex-matched controls.**

| | SS (n = 45) | Controls (n = 45) | | |
|---|---|---|---|---|
| | % (n) | % (n) | p-value | Odds ratio |
| Hyposthenuria | 35.71% (15) | 2.44% (1) | < 0.001 | 22 [3 – 958] |
| Glomerular hyperfiltration | 45.24% (19) | 21.43% (9) | 0.049 | 2.73 [0.89–8.61] |
| Normal glomerular filtration | 40.48% (17) | 52.38% (22) | NA | NA |
| Glomerular hypofiltration | 11.90% (5) | 23.81% (10) | 0.492 | NA |
| Renal insufficiency | 2.38% (1) | 2.38% (1) | 0.86 | NA |
| Microalbuminuria | 47.73% (21) | 8.89% (4) | < 0.001 | 9.36 [2.64–41.02] |
| Proteinuria | 37.21% (16) | 2.22% (1) | < 0.001 | 26.07 [3.5–1117.4] |
| Glomerular proteinuria | 0 | 0 | NA | NA |
| Tubular proteinuria | 39.02% (16) | 2.22% (1) | < 0.001 | 28.2 [3.8–1206.7] |
| Microglucosuria | 25% (11) | 6.67% (3) | 0.018 | 4.67 [1.09–27.69] |

SS: sickle cell anemia patients without diabetes. Controls: subjects without sickle cell anemia, sickle cell trait and diabetes. NA: not applicable.

with a broad spectrum of abnormal biological indices of kidney dysfunction. This was, to some extent, unexpected as most patients had the relatively favorable Senegal haplotype. This research will contribute in filling the gap in the investigation of kidney dysfunction in an African cohort. It has emphasized the need to improve prevention and care for all SCD patients in Africa regardless of their genetic and regional background.

Controls were recruited at random without an attempt to discriminate between age and sex with the cases. This recruitment protocol was due to three reasons. First, it was due to the difficulties associated with the recruitment of healthy controls in our low and middle income country (LMIC). Healthy individuals are not used to doing health check-ups as the latter are believed to be expensive. The few people benefiting from health check-ups are employees of a few companies or those applying for substantial loans from the bank. Companies or banks deal directly with clinics or laboratories in the private sector that are not involved in research activities. In our deserted blood centers, we avoid asking volunteer blood donors to participate in study protocols for fear of frightening them away. It was thus necessary to collaborate with associations of medical students to organize free medical campaigns in two popular suburbs of Dakar with logistic support from the local public authorities. The healthy individuals coming to consult were asked to participate in the study on the basis of an informed consent. In this context, it is difficult to sort the controls to be included. Also, the recruitment protocol was necessary to mainly define a cut-off for microglucosuria. For this purpose, it should be computed on the 5th and 95th glucosuria percentile of at least 120 Senegalese people recruited at random who, supposedly were healthy individuals sharing the same genetic background (had the same ethnic origins) and lived in the same environmental conditions as the sickle cell patients [16, 17]. This exercise was essential to establishing the values of the glucosuria tailored to our context, which could be considered as pathological in the absence of a cut-off of a consensus as it existed for the albuminuria. Thirdly, the recruitment protocol was supposed to help us to establish the prevalence of reliable data of kidney dysfunction in both SCA patients and the controls. If we had, since the beginning, used matching in lieu of open recruitment, it would have skewed, distorted any kidney dysfunction prevalence as a result. It is for such reasons that we have conducted a two-step comparison: we first compared the SCA patients with the controls without paying attention to age and sex and in a second step, we have compared the SCA patients and their paired controls' age and sex.

The recruitment protocol explains the age and sex differences between the two groups and may also have had an impact on the comparative analysis results of the biological indices in "Tables 3 and 4' of the first comparison. As the controls were significantly older than the cases, the proportion of kidney dysfunction should be higher in the control than in the case group because of the deleterious effects of aging on the kidneys. But, kidney dysfunction was significantly more common among the sickle cell patients than among the controls. To verify whether the differences noted in this first comparison did not result from a possible bias introduced by the case-control age and sex differences, we made a second comparison. This involved a subgroup of cases matched on age and sex to a subgroup of controls without any statistically significant difference regarding the body mass index. This second comparison involving a smaller sample confirmed the results obtained with the first comparison "Tables 3 and 5".

The study focused on biomarkers of functional impairment of the kidney. It is believed to be the first Senegalese study to assess the renal concentrating ability during SCA. The loss of renal concentrating ability is supposed to be the most common kidney dysfunction in SCA [18–20]. It results in a hyposthenuria which can be demonstrated using various methods [20–23]. In our study, refractometry has enabled us to find among SCA patients a median urine specific gravity (USG) of 1.012, i.e. 480 milliosmoles superimposable on the one reported for

an American population living with SCA of 1.0125 (502 milliosmoles) and a Jamaican one of 1.010 (400 milliosmoles) "Table 2"[20, 23]. Hyposthenuria was defined in our study by a USG lower or equal to the 5th percentile of the control USGs *i.e.* by a USG ≤ 1.010 which is equivalent to 400 milliosmoles according to the equation mOsm/kgH$_2$O = (USG -1,000) x 40,000 where mOsm/kgH$_2$O represents the urine osmolality and USG the urine specific gravity "Table 1"[24]. Yet, 400 milliosmoles represents the maximum concentrating ability of the cortical nephron and the minimum concentrating ability of the juxta-glomerular nephrons [22]. Hyposthenuria was found among 35.3% of the SCA patients of the series "Table 4". This should mean that these SCA patients supposedly lost their *vasa recta* which, in association with the juxta-medullar nephrons, ensure through the mechanism of counter-current multiplication, the concentration of urine above 400 milliosmoles [9, 22]. Losses of *vasa recta* by hyposthenuric SCA patients has already been demonstrated by microradioangiography [22]. It is believed to be caused by the fact that the medullar environment is hypertonic, hypoxic and acidic. It is therefore conducive to the sickled shape of the erythrocytes containing hemoglobin S. This may result in occlusions, ischemia and micro-infarcti which, in the long run cause the *vasa recta* to be destroyed (*i.e.*, *in fine* in the loss of the renal concentrating ability) among SCA patients [9, 18, 25, 26]. It appears therefore that determining regularly the urine specific gravity has all its importance in the monitoring of sickle cell anemia in that it gives us an insight into the degree of destruction of the *vasa recta* and allows for the use of a treatment which could slow down the progression of the loss of these vessels, the integrity of which is essential to maintain a normal renal ability to concentrate urine.

In addition, occlusion of the vasa recta is said to trigger the release of such vasodilator substances as prostaglandins that are supposed to cause disturbances in glomerular filtration in sickle cell patients [9, 25, 26]. Many studies, in Africa and elsewhere, have focused on the variations of the glomerular filtration rate (GFR) during the sickle cell disease since the GFR is deemed to be the best global index of the kidney function. Two different formulas were used to determine GFR because they are not interchangeable within an age group. The GFR median observed in the series' SCA patients is superimposable on the ones described earlier with some Senegalese (130 ml/min/1.73m$^2$), Malian (133 ml/min/1.73m$^2$), Cameroonian (135.1 ml/min/1.73m$^2$), Ghanaian (136.09 ml/min/1.73m$^2$), Jamaican (137 ml/min/1.73m$^2$) and American (133 ml/min/1.73m$^2$) (140.7 ml/min/1.73m$^2$) SCA patients [13, 23, 27–29]. It is higher than the one reported with SCA patients from India (104.47 ml/min/1.73 m$^2$) [30]. It looks as if an increased GFR is frequent among SCA patients with the exception of the ones from India. This difference might find an explanation in that India is one of the cradles of the Arab-Indian haplotype, considered to be the haplotype associated with the highest levels of HbF that blocks erythrocytes sickle shaping, a basic step in the physiopathology of sickle cell nephropathy.

The study has assessed the proportion of SCA patients with a glomerular hyperfiltration. For want of a threshold consensus, the glomerular hyperfiltration (GHF) was defined by a GFR > 140 ml/min/1.73m$^2$ regardless of age or gender, similar to the approach taken by some authors [30–32]. The prevalence of GHF among the series SCA patients is believed to be comparable to the ones reported with American SCA patients from Tennessee (47%), with SCA patients from Cameroon (49.5%), in the Sickle Cell Disease Center in Paris Tenon Hospital (51%), Subsaharan Africa and the French West Indies (53.1%) and Brazil (53%) [15, 33–36]. On the other hand, it is believed to be lower than the one described for the American SCA patients from Massachusetts (66%) but it is higher than those of Indian SCA patients (16%) [30, 37]. It appears that glomerular hyperfiltration is common with SCA patients. The differences between the reported prevalence could be attributed to variability in the methods of determining the levels of creatinine and GFR used by the various authors, for want of a threshold consensus on the definition of GHF that can sometimes be very low (GFR > 120 ml/min/

$1.73m^2$), to the age difference between the patients and to the influence of the haplotypes. With time, GHF might lead to renal insufficiency [38].

The prevalence of renal insufficiency, defined by a GFR < 60 ml/min/$1.73m^2$ was then assessed. The prevalence of renal insufficiency recorded with the SCA patients of the series was superimposable on the ones found with the American SCA patients of Massachussetts (4%) and of Tennessee (5.1%) and among SCA patients of Brazil (5.1%) and India (5.68%) [33, 36, 37]. This comparable prevalence observed in patients with different haplotypes and living in different geographic regions could mean both factors do not have a significant effect on the onset of renal insufficiency in SCA patients.

Eventually, of all the studied kidney functional alterations, only hyposthenuria and glomerular hyperfiltration appeared to be attributable to SCA. A theory to explain the occurrence of GHF in SCA has already been formulated [9, 25]. We, nevertheless, are proposing a new one. Glomerular hyperfiltration might result from a failed attempt to correct hyposthenuria using the *macula densa* cells. Indeed, at the physiological state, a reduced GFR following a fall of the blood pressure, causes a reduced tubular osmolality [39]. In response to the hypo-osmolality detected by their osmoreceptors, the *macula densa* cells trigger the tubulo-glomerular feedback mechanism and initiate hormonal regulation with the effect of increasing blood pressure and therefore GFR [39]. Among SCA patients, the *macula densa* cells of the juxta-glomerular nephrons trigger these two regulatory mechanisms of the glomerular blood pressure in response to the tubular hypo-osmolality in order to correct a hypothetical reduction of the GFR. This has, as a consequence a glomerular hyperfiltration as the origin of the tubular hypo-osmolality detected by their osmoreceptors is not a lower GFR but rather the juxtaglomerular nephrons' loss of their ability to concentrate urine, following a destruction of the *vasa recta* at the renal medullar level.

Just like with the functional alteration biomarkers, the study showed interest in the biomarkers of kidney structural damages notably in albuminuria, proteinuria and its glomerular and/or tubular origins as well as in glucosuria.

The albuminuria was assayed by an immunoturbidimetrical method using antibodies that help to specifically target albumin among urinary proteins. In the series controls, the 95th percentile of albuminuria (RACU) was 32.43 mg/g and is therefore supposed to be superimposable on the lower conventional limit that defines microalbuminuria (30 mg/g) "Table 1". Considering the conventional definition, albuminuria stage A2 or microalbuminuria's prevalence is comparable to the one you can find in a multicenter study of Michigan US children and/or teenage (46%), American adult (43,5%), Georgian American (42%), French from Subsaharan Africa and French West Indian (41.5%), Port Harcourt Nigerian (42.7%) and Brazilian (39%) patients with SCA [27, 34, 36, 38, 40, 41]. It is, on the other hand, higher than those reported with some Americans of Ohio (34%), Jamaicans (26%), Nigerians (26%), Saudi children and adolescents (28.9%), and Saudi adults (25%) living with SCD [20, 23, 42–44]. It is, however, lower than the one obtained with Cameroonian SCD patients (60.9%) [15]. As for albuminuria stage A3, it was only noted in our series in a ten-year-old girl with the SCA (a prevalence of 0.61% (1/163)). The albuminuria stage A3 or macroalbuminuria's prevalence was found to be comparable to that recorded among Ugandan living with the SCD (1.31%) [45]. It is, on the other hand, lower than the ones described for American SCD patients of Georgia (26%) and Ohio (6%) [20, 38]. The American study in Georgia has the highest albuminuria stage A3 prevalence because the Bantu haplotype, which is the acutest haplotype of the 5 (five) described haplotypes is widely represented in the study of the Georgian population. The large number of reported prevalence cases of pathological albuminuria is believed to illustrate a wide acceptance of this parameter as a biomarker for an early diagnosis of kidney dysfunction in the SCD. The variations of the prevalence between the countries and sometimes

within the same country reported by the various teams might be explained by the patients' age, the different methods for determining albuminuria, the patients' living environment and/or their genetic backgrounds. Concerning clinical significance, albuminuria stage A3 always points to a glomerulopathy but the glomerular and/or tubular origin of albuminuria stage A2 has not yet been clearly established with the SCD patients [38].

Proteinuria was assessed in the study. It was assayed using the pyrogallol–molybdate red method, which is a non-specific spectrocolorimetric method that allows for the simultaneous determination of all the proteins present in the urine, including albumin [46]. In a healthy subject, proteinuria (RPCU) may reach 180 mg/g [47]. The 95[th] percentile proteinuria in the study controls is superimposable on the upper limit of the RPCU since it reached 181.48 mg/g "Table 1". Proteinuria is deemed to be pathological when it exceeds 200 mg/g in the urine from one voiding [48]. So did the prevalence of pathological proteinuria, or simpler, of proteinuria reach among the SCA patients of the series a value superimposable on the ones reported for some Ghanaian (40.8%), Saudi (41%) and American children from Baltimore (41%), with SCA [29, 49, 50]. It should be worth noting that the American study assayed the proteinuria in all the urine testing negative by dipstick whereas the Ghanaian study only used the test trip to determine the proteinuria. The prevalence of the series' proteinuria, on the other hand is higher than the ones reported with the Ghanaian (2.8%), the Carolinas American (6.2% with the children and 12% with the adolescents), Kuwaiti (13.6%) and adult Nigerian (28) SCD patients [51–55]. The common ground for these 4 studies is that they have either used the test strip to determine the proteinuria or only measured the urinary proteins when the test strip screen was positive, with the exception of the Kuwaiti study. The lack of sensitiveness of the test strips exacerbated by the diluted urine resulting from the loss of the renal concentrating ability in these sickle cell patients explain the low prevalence noted in these studies. The Kuwaiti study has determined the 24-hour proteinuria with all the patients [53]. As a consequence, the low prevalence of the reported proteinuria in this study is basically due to the greatest representativeness in this region of the Arab-Indian haplotype. Proteinuria is thus a pertinent biomarker of kidney dysfunction in Senegalese living with SCA. Therefore, the evaluation of its glomerular or tubular origin on the basis of its albumin composition should prove particularly interesting. In this study, it is the RACU/RPCU ratio that has been used to situate the glomerular or tubular origin of the proteinuria. Proteinuria was, as expected, physiological (RPCU $\leq$ 200 mg/g) and mainly of tubular origin (RACU/ RPCU < 59%) with 95.27% of the controls. No pathological proteinuria (RPCU > 200 mg/g) was of glomerular origin (RACU/ RPCU $\geq$ 59%) either with the controls or with the sickle cell patients. On the other hand, some pathological proteinuria (RPCU > 200 mg/g) of tubular origin (RACU/ RPCU < 59%) were registered with 40.97% of the sickle cell patients and with 4.73% of the controls with a significant difference (p < 0.001) "Table 4". As a result, sickle cell anemia is only associated with tubular proteinuria. This was due to the fact that the sickle cell disease is associated with a chronic hemolytic anemia, as is witnessed by the rate of low hemoglobin of the SCA patients "Table 3". During hemolysis, the hemoglobin released in the plasma passes through the glomerular barrier [56]. It competes with the filtered proteins, notably with the albumin, on the megalin and cubulin receptors expressed by the cells of the proximal convoluted tubule (PCT) to be reabsorbed [56]. Thus, albumin and low molecular weight proteins can be found in the urine and cause albuminuria stage A2 and, more generally, a proteinuria of tubular origin [56]. In addition, the catabolism of hemoglobin, of the hem in particular, captured by the cells of the PCT causes the formation of toxic oxygenated free radicals for the tubular cells [57]. The cells so damaged might no longer play a part in the reabsorption of the proteins, as well as in the filtered glucose, promoting thereby the onset of the tubular proteinuria, albuminuria stage A2, even glucosuria.

To verify the presence of these renal tubular damages, we assessed a parameter we called microglucosuria or pauciglucosuria. This parameter was defined as a higher or equal glucosuria to the 95th percentile of the controls' RGCU (*i.e.*an RGCU $\geq$ 20 mg/g) but could not be detected by the test strips and which was not a consequence of hyperglycemia "Table 1". Thus, the prevalence of the microglucosuria for SCA patients is believed to be superimposable to the prevalence of the increase of the urinary low molecular weight proteins which are biomarkers of the injuries in the proximal convoluted tubule. It notably includes an 18% prevalence of hyper-β2-microglobulinemia reported with Turkish SCD patients and a 16% prevalence of an increase of retinol binding protein (RBP) with some American SCA adolescents of Baltimore (16%) [50, 58]. Microglucosuria is thus a biomarker of proximal convoluted tubular damages in sickle cell disease. It is also an argument for the tubular origin of both albuminuria stage A2 and proteinuria in SCD patients. Marsenic O et al. reported that 15% of their series SCA patients with a proteinuria showed concurrently an increase of RBP [50]. As for Sundaram et al., they noted a vigorous association between albuminuria and N-acetyl-glucosaminidase (NAG), a lysosomal enzyme released by the damaged cells of the proximal convoluted tubule [20].

This study presents some limitations. Indeed, it was impossible to assess whether the biomarkers were transiently or permanently disturbed since this was a case-control study. In addition, odds ratios, which are very high in some cases, should reflect the presence of confounding factors that only a multivariate analysis could remove. Moreover, the serologic assays of the *hepatitis B virus*, *Streptococcus pneumoniae or Schistosoma haematobium* had have not been carried out though these pathogens may cause the kidney dysfunction in some controls or patients with SCA. Another limitation is selection of control group, it is open to biais, especially sex and age one. At last, GFRs are presented together while formulas used to compute them was different between child and adults.

In conclusion, this study showed a relatively high proportion of SCA nephropathies among patients living with SCA in Senegal. The study highlights that hyposthenuria, glomerular hyperfiltration, albuminuria stage A2, tubular proteinuria and microglucosuria could be relevant biomarkers of sickle cell nephropathy. It has revealed a biomarker, microglucosuria, which could be used as well as the urinary albumin/total protein ratio in association with proteinuria for screening kidney dysfunction in sickle cell anemia patients. The study to identify anthropometric, clinico-biological, genetic and even environmental risk factors that predispose these biomarkers to disturbances will be necessary to be able to identify patients at-risk and allow early detection and the therapeutic management of sickle cell nephropathy.

## Materials and methods

The study protocol complied with the ethical guidelines of the Helsinki Declaration and was approved by the Research Ethics Committee from Dakar Cheikh Anta Diop University (0312/2018/CER/UCAD) and by the Faculty of Health Sciences Human Research Ethics Committee from University of Cape Town (HREC RE: 661/2015). Participation was subjected to the free and informed consent of subjects who were at least 18 years old and parents or guardians of those under 18 years.

This was a case-control study that included SCA patients without diabetes (SS) and controls with no detectable SCA, sickle cell trait and diabetes.

Patients with SCA were recruited in Dakar (Senegal) at the» National Blood Transfusion Center «Centre National de Transfusion Sanguine (CNTS)», the reference care center for adults with SCA; and the Ambulatory Care Unit for Children and Adolescents with Sickle Cell Disease «Unité de Soins Ambulatoires des enfants et adolescents atteints de Drépanocytose (USAD)» located at the Albert Royer National University Children's Hospital « Centre

Hospitalier National d'Enfants Albert Royer (CHEAR) », the largest care unit for children and adolescents with SCA in Senegal. The control participants were recruited at random during two campaigns of free medical consultations organized in two suburbs of Dakar. Patients with SCA were included in the study if they were already enrolled in the sickle cell adult or children cohort, at least 4 years of age, at a routine fasting visit, and in steady state health. SCA patients or controls were defined as adults when they were more than 20 years old. They were classified as children when they were under 20 years old. The exclusion criteria included those in a pain crisis and/or with diabetes. Samples from control participants were collected when they were apparently healthy and at least 4 years old. Control participants were excluded from the study when their hemoglobin solubility test was positive and their *HBB* genotype was $\beta^S/\beta^A$ and/or their fasting blood sugar $\geq$ 1.26 g/l.

Venous blood and random midstream urine specimens were collected after at least 8 hours of fasting. The assessment of biological indices was conducted at the Clinical Biochemistry Laboratory of Albert Royer National University Children's Hospital of Dakar (CHEAR). Quantitative assay of hemoglobin by sodium lauryl sulfate, a cyanide-free reagent, was performed using Sysmex XT-4000i (Sysmex Corporation, Kobe, Japan). Using a Mindray-BS-380 clinical biochemistry analyzer (Mindray, Créteil, France) and Biosystems reagents (Biosystems reagents & instruments, Barcelone, Espagne), the following parameters were analyzed using spectrocolorimetry: glycemia and glucosuria by glucose oxidase / peroxidase method, blood urea nitrogen (BUN) by urease / Berthelot reagent method, serum creatinine and urine creatinine by creatininase / creatinase / sarcosine oxidase / peroxidase enzymatic method with standardization to isotope dilution mass spectrometry, proteinuria by pyrogallol red molybdate, albuminuria by immunoturbidimetric method using a specific anti-human albumin antibodies. Glucose was also screened in urine by glucose oxidase / peroxidase method using test strips (nal von minden GmbH, Regensburg, Germany). Urine specific gravity was measured using an Atago-SPR-T2 refractometer (Atago, Saitama, Japon). GFR was computed using Schwartz's formula in children and adolescents, and Chronic Kidney Disease—EPIdemiology (CKD-EPI) equation in adults [59, 60].

Proteinuria and albuminuria were normalized with urine creatinine and expressed as a ratio. Thus, proteinuria was expressed as a urinary protein to creatinine ratio (UPCR) and albuminuria as a urinary albumin to creatinine ratio (UACR). All two ratios were expressed as mg of protein or albumin per g of urine creatinine (mg/g). UPCR was defined as pathological proteinuria (UPCR > 200 mg/g [22.6 mg/mmol]) or physiological proteinuria (UPCR $\leq$ 200 mg/g [22.6 mg/mmol]). The urinary albumin / total protein ratio (UACR/UPCR) indicated the origin of proteinuria which was qualified as glomerular (UACR/UPCR $\geq$ 59%) or tubular (UACR/UPCR < 59%) [47, 61]. Thus, for example, normal glomerular proteinuria was defined as UPCR $\leq$ 200 mg/g (22.6 mg/mmol) with UACR / UPCR $\geq$ 59% while tubular pathologic proteinuria was defined as UPCR > 200 mg/g (22,6 mg/mmol) with UACR/ UPCR < 59%. UACR was defined as albuminuria stage A1 (UACR < 30 mg/g [3.39 mg/ mmol]), albuminuria stage A2 (30 mg/g [3.39 mg/mmol] $\leq$ UACR < 300 mg/g [33.9 mg/ mmol]) or albuminuria stage A3 (UACR $\geq$ 300 mg/g [33.9 mg/mmol]). Microglucosuria was defined as glucosuria which is not a consequence of hyperglycemia and which might not be detectable by urine test strips that generally do not detect glucosuria below 50 mg/dl (2.775 mmol/l) but which is quantifiable by glucose oxidase / peroxidase method which can determine glucosuria 200 times lower (0.23 mg/dl [0.013 mmol/l]) according to the manufacturers of the reagents used in our study. Glucosuria was normalized by computing the ratio of glucosuria (mg/dl) to urine creatinine (g/dl), abbreviated UGCR, expressed in mg/g (x 0.625 μmol/ mmol). Glucosuria greater than or equal to the 95[th] percentile of the UGCR in the control group was considered to be microglucosuria or pauciglucosuria. Hyposthenuria qualified an

USG $\leq 5^{th}$ percentile of the USG observed in the control group. Glomerular hyperfiltration (GHF) was defined by GFR > 140 ml/min/1.73m$^2$, normal glomerular filtration by $90 \leq$ GFR $\leq 140$ ml/min/1.73 m$^2$, glomerular hypofiltration by $60 \leq$ GFR < 90 ml/min/1.73 m$^2$ and renal insufficiency by GFR < 60 ml/min/1.73m$^2$.

DNA was extracted from peripheral blood at the Clinical Biochemistry Laboratory of Albert Royer National University Children's Hospital of Dakar (CHEAR) using Puregene Blood Kit (Qiagen, Hilden, Germany). Molecular confirmation of SCA was performed at the Division of Human Genetics, Faculty for Health Sciences, University of Cape Town using restriction fragment length polymorphism (RFLP) with the same materials and protocols previously described [15]. Molecular analysis to identify the presence of the sickle mutation was carried out by polymerase chain reaction (PCR) to amplify a 770 bp segment of *HBB*, followed by DdeI restriction analysis of the PCR product [15]. Genotyping for the *XmnI*-rs7482144 was performed using the iPLEX Gold Sequenom Mass Genotyping Array (Inqaba Biotec, Pretoria, South Africa).

Regarding matching, we proceeded in two steps. In the first step, we compared cases and controls without considering age and sex as matching parameters. In second step, we selected a subgroup of cases who matched on age and sex with a subgroup of controls without any statistical significant difference regarding body mass index. Then these two subgroups were compared. In both steps, cases and controls was ethnic matched: all subjects included in both groups were sub-Saharan African black people. Descriptive statistics was used for anthropometric and biological variables (median, minimum, maximum, $5^{th}$ and $95^{th}$ percentiles), for both cases and controls. In addition, the Wilcoxon-Mann-Whitney test was used to compare the means, for quantitative variables, between cases and unmatched controls, and between cases and controls matched on age and sex. Relevant quantitative parameters of nephropathy were transformed into categorical variables. The comparison of the prevalence of biomarkers disturbances was carried out using the χ2 test between unmatched cases and controls and then between cases and controls matched on age and sex. When an association was statistically established, the odds ratio (OR) was then calculated. The significance level for the tests was set at p < 0.05. Statistical analysis was carried out using STATA version 14.0.370 for Windows TM (Stata Corp Inc., College Station, Texas, USA).

## Supporting information

**S1 File.**
(PDF)

**S2 File.**
(XLS)

**S3 File.**
(PDF)

**S1 Data.**
(XLS)

**S2 Data.**
(DOCX)

## Author Contributions

**Conceptualization:** El Hadji Malick Ndour, Khuthala Mnika, Victoria Nembaware, Gaston Kuzamunu Mazandu, Aynina Cissé, Ambroise Wonkam, Philomène Lopez Sall.

**Data curation:** El Hadji Malick Ndour, Khuthala Mnika, Ambroise Wonkam.

**Formal analysis:** El Hadji Malick Ndour, Khuthala Mnika, Gaston Kuzamunu Mazandu, Aynina Cissé, Ambroise Wonkam, Philomène Lopez Sall.

**Funding acquisition:** El Hadji Malick Ndour, Aynina Cissé, Ambroise Wonkam, Philomène Lopez Sall.

**Investigation:** El Hadji Malick Ndour, Khuthala Mnika, Aynina Cissé, Ambroise Wonkam, Philomène Lopez Sall.

**Methodology:** El Hadji Malick Ndour, Khuthala Mnika, Gaston Kuzamunu Mazandu, Aynina Cissé, Ambroise Wonkam, Philomène Lopez Sall.

**Project administration:** El Hadji Malick Ndour, Victoria Nembaware, Ambroise Wonkam.

**Resources:** El Hadji Malick Ndour, Moussa Seck, Indou Dème Ly, Victoria Nembaware, Rokhaya Dione, Aliou Abdoulaye Ndongo, Jean Pascal Demba Diop, Saliou Diop, Ibrahima Diagne, Ambroise Wonkam, Philomène Lopez Sall.

**Software:** El Hadji Malick Ndour, Ambroise Wonkam.

**Supervision:** Khuthala Mnika, Aynina Cissé, Ambroise Wonkam, Philomène Lopez Sall.

**Validation:** Khuthala Mnika, Gaston Kuzamunu Mazandu, Aynina Cissé, Ambroise Wonkam, Philomène Lopez Sall.

**Writing – original draft:** El Hadji Malick Ndour.

**Writing – review & editing:** El Hadji Malick Ndour, Khuthala Mnika, Fatou Guèye Tall, Moussa Seck, Indou Dème Ly, Victoria Nembaware, Gaston Kuzamunu Mazandu, Hélène Ange Thérèse Sagna Bassène, Rokhaya Dione, Aliou Abdoulaye Ndongo, Jean Pascal Demba Diop, Nènè Oumou Kesso Barry, Moustapha Djité, Rokhaya Ndiaye Diallo, Papa Madièye Guèye, Saliou Diop, Ibrahima Diagne, Aynina Cissé, Ambroise Wonkam, Philomène Lopez Sall.

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
