## [Decision Letter · Decision Letter 0]

18 Aug 2021

PONE-D-21-14690

BIOMARKERS OF SICKLE CELL NEPHROPATHY IN SENEGAL

PLOS ONE

Dear Dr. NDOUR,

Thank you for submitting your manuscript to PLOS ONE. After careful consideration, we feel that it has merit but does not fully meet PLOS ONE’s publication criteria as it currently stands. Therefore, we invite you to submit a revised version of the manuscript that addresses the points raised during the review process. Please make some major revisions and explanation to some of the point raised by the reviewers, we will review again once changes have been made and I look forward to revised version.

We look forward to receiving your revised manuscript.

Kind regards,

Bhagwan Dass, MD

Academic Editor

PLOS ONE

Journal Requirements:

2. Please amend the manuscript submission data (via Edit Submission) to include author Khuthala Mnika, Fatou Guèye Tall1, Moussa SECK, Indou Dème Ly, Victoria Nembaware, Gaston Kuzamunu Mazandu, Hélène Ange Thérèse Sagna-Bassène, Rokhaya Dione, Aliou Abdoulaye Ndongo, Jean Pascal Demba Diop, Nènè Oumou kesso Barry, Moustapha Djité Rokhaya Ndiaye Diallo, Papa Madièye Guèye, Saliou Diop, Ibrahima Diagne, Aynina Cissé1, Ambroise Wonkam and Philomène Lopez Sall. 

Reviewers' comments:

Reviewer's Responses to Questions

**Comments to the Author**

1. Is the manuscript technically sound, and do the data support the conclusions?

Reviewer #1: Yes

Reviewer #2: Partly

2. Has the statistical analysis been performed appropriately and rigorously? 

Reviewer #1: Yes

Reviewer #2: I Don't Know

3. Have the authors made all data underlying the findings in their manuscript fully available?

Reviewer #1: Yes

Reviewer #2: No

4. Is the manuscript presented in an intelligible fashion and written in standard English?

Reviewer #1: Yes

Reviewer #2: No

5. Review Comments to the Author

Reviewer #1: This is a well designed studies answering relevant questions for the chosen study population with potentially wider implications for the management of patients with SCD worldwide.

The statistics, presentation of results and discussion are sound and logical.

I have made minor comments on the attached pdf.

The main recommendation is for the authors to discuss a bit more the age difference between thecases and controls, whether that might have impacted findings and if so how

Reviewer #2: Review assignment for PONE-D-21-14690

BIOMARKERS OF SICKLE CELL NEPHROPATHY IN SENEGAL

PLOS ONE

Sickle cell nephropathy is a common complication of SCD, which is one of the most frequent monogenic disorders worldwide. As Senegal is a country with high prevalence of the disease, studying the biomarkers of sickle cell nephropathy in Senegal is a relevant topic. However, there are major concerns regarding the methodology. Additionally, the results could have been reported more efficiently and in a less ambiguous way.

Major concerns

1 – Regarding the statistical methods, the authors do not clearly explain how the controls were selected. Although restriction to an ethno-linguistic group may be relevant, there is a lack of information about how the matching was performed.

2 – The definition presented for sickle cell disease is actually the definition for sickle cell anemia (patients whose phenotype is usually the most severe because of being homozygous for the beta S mutation).

3 – Please consider merging table 1 and 2 into a single table with columns for SCA patients, controls, and p-values. Keeping the number of observations for each variable is appropriate. However, median and interquartile range would be preferable to median (min-max) and 5-95th percentiles).

Minor concerns

1 – The same keyword should be used throughout the paper when referring to “kidney dysfunction” for consistency. Avoid using dysfunction, impairment, abnormalities, etc., to refer to the same concept.

2 – The authors could consider using “without diabetes” instead of “free of diabetes”.

3 – The non-normally distributed continuous variables usually are presented as median (interquartile range). The interquartile range is the range from the 25th to 75th percentile, not the range from the 5th to 95th percentile which is presented by the authors.

4 – The authors should uniformize the way p-values are presented. For example, “p-value <0.001” instead of “p-value <10^-3” consistently.

5 – The sex distribution of the study sample should be presented as absolute and relative frequencies of the most common category/sex, instead of the female-to-male ratios.

6 – In line with the KDIGO guidelines, the albuminuria stages should be used instead of the older terms “microalbuminuria” and “macroalbuminuria”. Also, the term “chronic kidney disease” should be used instead “chronic renal insufficiency”.

7 - Glycosuria is a known marker of tubular dysfunction and the authors appropriately stress the importance of monitoring glycosuria as an early biomarker of sickle cell nephropathy. However, the term “microglucosuria” sounds inappropriate as it focus on the capacity of the laboratorial test to detect glycose in the urine.

8 – A high urine protein-to-creatinine ratio (UPCR) with normal urine albumin-to-creatinine ratio (UACR) is common in sickle cell nephropathy as a mechanism of tubular proteinuria is expected. The authors define tubular proteinuria as a UPCR-UACR ratio > 59%. The reason to use this cut-off should have been discussed. Please provide a reference for this.

9 – In table 4 avoid showing the observations for all categories of each variable as it is redundant. Please show the observations for the category of interest of each variable (for example, show the frequency of urinary albumin to creatinine ratio >= 30 mg/g). Also, the GFR categories could be presented in a decrescent order (e.g., > 140; 90-140; 60-90; <60).

6. PLOS authors have the option to publish the peer review history of their article (what does this mean?). If published, this will include your full peer review and any attached files.

Reviewer #1: **Yes: **Dr Leonard Ebah

Reviewer #2: No

---

## [Author Response · Author response to Decision Letter 0]

27 Aug 2021

We thank the editor and reviewers for their helpful comments. The manuscript has been revised to address their concerns, with additions and changes highlighted in blue and support information.

---

## [Decision Letter · Decision Letter 1]

19 Jan 2022

PONE-D-21-14690R1BIOMARKERS OF SICKLE CELL NEPHROPATHY IN SENEGALPLOS ONE

Dear Dr. NDOUR,

Thank you for submitting your manuscript to PLOS ONE. After careful consideration, we feel that it has merit but does not fully meet PLOS ONE’s publication criteria as it currently stands. Therefore, we invite you to submit a revised version of the manuscript that addresses the points raised during the review process.

We look forward to receiving your revised manuscript.

Kind regards,

Laurent Metzinger

Academic Editor

PLOS ONE

Reviewers' comments:

Reviewer's Responses to Questions

**Comments to the Author**

1. If the authors have adequately addressed your comments raised in a previous round of review and you feel that this manuscript is now acceptable for publication, you may indicate that here to bypass the “Comments to the Author” section, enter your conflict of interest statement in the “Confidential to Editor” section, and submit your "Accept" recommendation.

Reviewer #3: All comments have been addressed

2. Is the manuscript technically sound, and do the data support the conclusions?

Reviewer #3: Yes

3. Has the statistical analysis been performed appropriately and rigorously? 

Reviewer #3: Yes

4. Have the authors made all data underlying the findings in their manuscript fully available?

Reviewer #3: Yes

5. Is the manuscript presented in an intelligible fashion and written in standard English?

Reviewer #3: Yes

6. Review Comments to the Author

Reviewer #3: Ndour et la present an original article entitled "BIOMARKERS OF SICKLE CELL NEPHROPATHY IN SENEGAL"

The study is well designed and relevant asq it focuses on a specific population.

The statistics, presentation of results and discussion are mostly OK.

I would advise to justify the age difference between the patients and controls.

Also in discussion, compare with more depth to other relevant similar papers in other populations.

7. PLOS authors have the option to publish the peer review history of their article (what does this mean?). If published, this will include your full peer review and any attached files.

Reviewer #3: No

---

## [Author Response · Author response to Decision Letter 1]

7 Mar 2022

Thank you for your recommandations. They have helped us to improve the discussion section of the paper and to deposit in protocols.io the laboratory protocol that describes how microglucosuria is determined in patients living with sickle cell disease in our paper.

---

## [Decision Letter · Decision Letter 2]

3 Jun 2022

PONE-D-21-14690R2BIOMARKERS OF SICKLE CELL NEPHROPATHY IN SENEGALPLOS ONE

Dear Dr. Ndour,

Thank you for submitting your manuscript to PLOS ONE. After careful consideration, we feel that it has merit but does not fully meet PLOS ONE’s publication criteria as it currently stands. Therefore, we invite you to submit a revised version of the manuscript that addresses the points raised during the review process.

We look forward to receiving your revised manuscript.

Kind regards,

Donovan Anthony McGrowder, PhD., MA., MSc

Academic Editor

PLOS ONE

Additional Editor Comments (if provided):

Dear Dr. Ndour,

Your manuscript “BIOMARKERS OF SICKLE CELL NEPHROPATHY IN SENEGAL”” has been assessed by our reviewers. They have raised a number of points which we believe would improve the manuscript and may allow a revised version to be published in PLOS ONE. Their reports, together with any other comments, are below.

If you are able to fully address these points, we would encourage you to submit a revised manuscript to PLOS ONE.

Reviewers' comments:

Reviewer's Responses to Questions

**Comments to the Author**

1. If the authors have adequately addressed your comments raised in a previous round of review and you feel that this manuscript is now acceptable for publication, you may indicate that here to bypass the “Comments to the Author” section, enter your conflict of interest statement in the “Confidential to Editor” section, and submit your "Accept" recommendation.

Reviewer #4: (No Response)

Reviewer #5: (No Response)

Reviewer #6: (No Response)

2. Is the manuscript technically sound, and do the data support the conclusions?

Reviewer #4: Yes

Reviewer #5: Partly

Reviewer #6: Partly

3. Has the statistical analysis been performed appropriately and rigorously? 

Reviewer #4: No

Reviewer #5: No

Reviewer #6: Yes

4. Have the authors made all data underlying the findings in their manuscript fully available?

Reviewer #4: Yes

Reviewer #5: (No Response)

Reviewer #6: Yes

5. Is the manuscript presented in an intelligible fashion and written in standard English?

Reviewer #4: Yes

Reviewer #5: Yes

Reviewer #6: Yes

6. Review Comments to the Author

Reviewer #4: El Hadji Malick Ndour et al performed case-control study to identify the candidate biomarkers for sickle cell nephropathy in sickle cell anemia from Senegal population. A total of 163 patients with SCA and 177 ethnic matched controls were enrolled. The higher prevalence of hyposthenuria, glomerular hyperfiltration, microalbuminuria, proteinuria, tubular proteinuria, and microglucoruia were observed in SCA patients compared with controls. Although the study is rather descriptive, the study to investigate the prevalence of renal dysfunction in SCA patients is unique and interesting. However, there are several concerns as follow.

Major comments

1. In abstract line 5, what is the definition and diagnostic criteria for “sickle cell nephropathy”? The authors should apply the diagnostic criteria for “sickle cell nephropathy” in the SCA patients. In addition, how did the authors exclude other kidney diseases in SCA and control groups?

2. In line 63-65 Table 1 and 2, were the patients with diabetes excluded? Were there no patients with blood glucose levels more than 170-180 mg/dL?

3. In lines 91 and 272-274, the authors should describe the definition of glomerular hyperfiltration.

4. In line 92, the authors should describe the definition of chronic kidney disease (CKD).

5. In line 93, did the authors measure tubular injury markers such as β2-microglobulin?

6. In lines 306-309, how did the authors statistically match the age and sex? Did the authors use the propensity score matching?

7. If the authors are able to make the diagnosis “sickle cell nephropathy” according to the diagnostic criteria, the authors investigate the sensitivity, specificity and AUC of cut-off points of hyposthenuria, glomerular hyperfiltration, microalbuminuria, proteinuria, tubular proteinuria, and microglucoruia for the diagnosis of “sickle cell nephropathy”.

Reviewer #5: The authors showed a relatively high proportion of SCA nephropathy among patients living with SCA in Senegal, and several parameters were suggested as prevalent biomarkers of nephropathy. This article seems to be refined after appropriate revision, and the contents is interesting. However, I regret to realize several problems of the article.

1. This is a critical problem. The authors ignored the effect of the skeletal muscle volume on serum creatinine levels and urinary creatinine excretion. The skeletal muscle volume was known to decrease in patients with SCA in both child and adult (Barden et al. Am J Clin Nutr. 2002; 76: 218-25., Ravelojaona et al. Am J Pathol. 2015; 185: 1448-56.). Both serum creatinine levels and urinary creatinine excretion decrease in patient with lower skeletal muscle volume. Hence, eGFR, UACR and UPCR would be overestimated in patients with SCA. Actually, although statistical difference for BMI was not observed among age- and sex-matched patients, it strongly tended to lower in patients with SCA (Table 3: p=0.079). BMI would decrease in patients with lower skeletal muscle volume. Because patient number was relatively small, hence beta error for BMI should be considered.

2. This is a second critical problem. The timing to collect urine for urinary analysis should be described. Because amount of water drinking or food intake can affect USG, especially in non-early morning urine.

3. Did the authors evaluate urinary volume, amount of drinking and serum sodium concentration? These parameters would be different in patients with SCA because impairment of urinary concentration was observed.

4. Which one was the most important biomarker for detect SCA nephropathy? Was there any difference between early and late phase? Which biomarker was associated with poor renal prognosis?

5. I may be wrong but could you confirm line 172 and 185? Was it "Table 5" instead of "Table 4"? Because the authors described “Table 5” in both sentences, but the values for the prevalence of GHF and CKD, which were described in line 171 and 183, were same to the values described in Table 4.

6. I may be wrong but could you confirm the value for blood pressure? I think “regular” blood pressure is about 120/70 mmHg, but the described value seems one digit missing.

Reviewer #6: Dear authors,

Thank you for this submission.

My comment are given below.

Abstract

The terms creatininemia nad creatinuria do not sound right as everyone has creatinine in blood and urine. They are not used widely. Please revise term with known, widely used and accepted ones. This is valid for these terms through all manuscript. My suggestion is creatinine and urine creatinine.

Introduction

Details about sickle cell disease are very important but as the topic is sickle cell nephropathy they could be summarized to highlight nephropathy more.

Results

I think there is no need to give 5-95 percentiles at table 1 and 2. Please revise them and unite table 1 and 2 which would make them easily seen together and comparable. In addition, a column could be added to this table for p values which would make table 3 unnecessary as well. There is no need to define controls as AA non-DT as details of them are given at the text. It may be refereed just as control group. In summary, in my opinion, a table for all participants’ values with p values and another table for just sex and age matched controls containing all parameters would be more suitable. P values should be present for all parameters given at tables, not for some parameters.

Blood pressures are written to be in mmHg but there should a mistake at numbers. They are so small and I think are given in cmHg. Please check them.

Details between line 75-80 are methodological ones so they should be removed to methods section. There should be no reference given at results section.

Please revise sentences between lines 91 and 103. Adding ‘table…’ to the end of long sentences are not grammatically correct. Parameters could be given one by one and detailed according to the groups, sickle cell and control, and p value could be given at last. Moreover, there should be consistency between definition of parameters at the text and at the table. If you mention about albuminuria stage 2 at the text, a reader must see this parameter when looking to the table easily. Revise text and tablet o make them parallel to each other. Instead of stage 2 albuminuria, microalbuminuria could be used if you prefer.

When I read results section, it made my mind confused rather than being aware of differences between patients and controls. It should be more precise.

Discussion

This section should not repeat findings. To refer a finding, a short sentence referring findings is enough and there should be some comment or speculation about it besides comparison with other similar studies. Please try to remove repetitions regarding finding as much as possible.

GFR was found by different formula in children and adults. Is it correct to put GFRs got by two different formula and present them together? I think it is not correct. The percentage of children is not given. You should give this detail. If you would like to present GFRs together, you should add this a limitation. Another limitation is the selection of control group. It is open to bias, especially sex and age matched one. It should be also added as a limitation. The details of GFR measurements should be transferred to methods section. At discussion, the reason for this and its possible effects to th

---

## [Author Response · Author response to Decision Letter 2]

26 Jul 2022

We thank reviewers for their helpful observations, questions, comments and recommantions. 

We have given our response in the rebuttal letter.

---

## [Editor Report · Decision Letter 3]

16 Aug 2022

BIOMARKERS OF SICKLE CELL NEPHROPATHY IN SENEGAL

PONE-D-21-14690R3

Dear Dr. Ndour,

We’re pleased to inform you that your manuscript has been judged scientifically suitable for publication and will be formally accepted for publication once it meets all outstanding technical requirements.

Kind regards,

Donovan Anthony McGrowder, PhD., MA., MSc

Academic Editor

PLOS ONE

Dear Dr. Ndour,

The manuscript was revised in accordance with the reviewers’ comments and is provisionally accepted pending final checks for formatting and technical requirements.

Regards,

Dr. Donovan McGrowder (Academic Editor)<o:p></o:p>

---

## [Editor Report · Acceptance letter]

9 Nov 2022

PONE-D-21-14690R3 

Biomarkers of Sickle Cell Nephropathy in Senegal 

Dear Dr. Ndour:

I'm pleased to inform you that your manuscript has been deemed suitable for publication in PLOS ONE. Congratulations! Your manuscript is now with our production department. 

Kind regards, 

on behalf of

Dr. Donovan Anthony McGrowder 

Academic Editor

PLOS ONE